

# Comment on Pescott & Jitlal 2020: Failure to account for measurement error undermines their conclusion of a weak impact of nitrogen deposition on plant species richness

Simon M. Smart[1], Carly J. Stevens[2], Sam J. Tomlinson[1], Lindsay C. Maskell[3] and Peter A. Henrys[3]

[1] Centre for Ecology & Hydrology Lancaster, Lancaster, United Kingdom
[2] Lancaster Environment Centre, Lancaster University, Lancaster, United Kingdom
[3] Land Use, Centre for Ecology & Hydrology Lancaster, Lancaster, United Kingdom

Corresponding author
Simon M. Smart, ssma@ceh.ac.uk

## ABSTRACT

Estimation of the impacts of atmospheric nitrogen (N) deposition on ecosystems and biodiversity is a research imperative. Analyses of large-scale spatial gradients, where an observed response is correlated with measured or modelled deposition, have been an important source of evidence. A number of problems beset this approach. For example, if responses are spatially aggregated then treating each location as statistically independent can lead to biased confidence intervals and a greater probably of false positive results. Using methods that account for residual spatial autocorrelation, *Pescott & Jitlal (2020)* re-analysed two large-scale spatial gradient datasets from Britain where modelled N deposition at $5 \times 5$ km resolution had been previously correlated with species richness in small quadrats. They found that N deposition effects were weaker than previously demonstrated leading them to conclude that "*previous estimates of Ndep impacts on richness from space-for-time substitution studies are likely to have been over-estimated*". We use a simulation study to show that their conclusion is unreliable despite them recognising that an influential fraction of the residual spatially structured variation could itself be attributable to N deposition. This arises because the covariate used was modelled N deposition at $5 \times 5$ km resolution leaving open the possibility that measured or modelled N deposition at finer resolutions could explain more variance in the response. Explicitly treating this as spatially auto-correlated error ignores this possibility and leads directly to their unreliable conclusion. We further demonstrate the plausibility of this scenario by showing that significant variation in N deposition at the 1 km square resolution is indeed averaged at $5 \times 5$ km resolution. Further analyses are required to explore whether estimation of the size of the N deposition effect on plant species richness and other measures of biodiversity is indeed dependent on the accuracy and hence measurement error of the N deposition covariate. Until then the conclusions of *Pescott & Jitlal (2020)* should be considered premature.

## INTRODUCTION

Atmospheric nitrogen deposition is one of a number of chronic pressures that arise from human activity (*Ackerman, Millet & Chen, 2018*; *Sala et al., 2000*). Since nitrogen is an essential macronutrient that is limiting in many ecosystems, unnaturally high levels of deposited N are expected to cause a range of ecological effects (*Stevens et al., 2011*; *RoTAP, 2012*; *Phoenix et al., 2012*). These effects are also modified by factors such as livestock grazing (*Van der Wal et al., 2003*), historical sulphur deposition (*RoTAP, 2012*; *Rose et al., 2016*), soil pH (*Van Den Berg et al., 2011*; *Van Den Berg et al., 2005*), P limitation (*Rowe, Smart & Emmett, 2014*) and species identity (*Van Den Berg et al., 2005*; *Sheppard et al., 2014*).

Measuring the impacts of excess N on ecosystem processes and biodiversity involves a range of approaches including large-scale gradient analyses (summarised for Great Britain in *RoTAP, 2012*). These are challenging analyses to carry out. Unlike an experimental manipulation, demonstrating a causal link between N deposition and an ecological response is complicated by varying levels of inaccuracy in the N deposition estimates and response data and also where interacting or confounding variables, such as those listed above, are unavailable or measured or modelled at coarser resolutions than the response data. A strength of these spatial gradient studies is their realism but at the cost of uncertainty in attributing cause to effect (*Smart et al., 2012*).

*Pescott & Jitlal (2020)* (hereafter P&J20) reanalysed two of the datasets from two large-scale spatial gradient analyses. These are *Stevens et al. (2004)* and *Maskell et al. (2010)*. Following P&J20 we refer to these as MEA10 and SEA04. Their reanalysis led P&J20 to conclude that N deposition effects on plant species richness were revealed to be small and ambiguous and their previous importance had been overstated. A wide range of evidence has led to major policy interest in addressing the causes and consequences of excess N inputs to ecosystems of which atmospheric sources are a significant fraction. Therefore the conclusion of P&J20 requires careful scrutiny since it appears to cast doubt on the importance of N deposition as a driver of ecological change.

### The conclusion that "spatial Bayesian linear models indicate small and ambiguous effects on species richness" is not proven

We suspected that a failure to investigate the role of measurement error in the modelled N deposition estimates led them to falsely infer a weak effect when they applied a model that accounted for spatial autocorrelation. We report a simulation study that supports our case.

In their reanalysis, P&J20 used exactly the same N deposition estimates at $5 \times 5$ km resolution as used by MEA10 and SEA04. The response data was observed plant species count in $2 \times 2$ m quadrats. Using the same $5 \times 5$ km modelled N deposition values ensured that their results should only differ from those of MEA10 and SEA04 because of differences in modelling methods. We show below that the influence of measurement error associated with the modelled N deposition covariate can explain the weakening of the detected N deposition effect when they include a spatial field that accounts for the non-independence of the observed responses. We should be clear that our objection is not with the technical

implementation of their spatial model, which as far as we can tell seems correct. It is the conclusion they draw from their results that we believe is unreliable.

P&J20 applied a model to the two datasets that accounted for possible spatial dependence across the sample of species richness values in MEA10 and SEA04. If present and not taken into account this could inflate the effect attributable to N deposition. We agree that this is potentially a valid concern should there remain any spatial autocorrelation in the residuals arising from the models of MEA10 and SEA04. However, P&J20 did not explore analytically the potential influence of measurement error on their results. By measurement error we mean the deviation between the species richness response and the modelled covariate that would be reduced if the modelled value equalled, or was a more accurate approximation of, the true N deposition value. Assuming a true underlying relationship between N deposition and species richness, the regression parameter describing this relationship would be more accurate (nearer the truth) and more precise (less uncertain) if this deviation was reduced. Coarsening the resolution of the N deposition estimates as well as process-based deficiencies in the N deposition model will decrease the slope of the relationship between the response and the covariate. For these reasons we would expect an upper limit to the explanatory power of the modelled N deposition at $5 \times 5$ km. The important point here is that this limitation reflects sources of inaccuracy in the explanatory variable. This means that some of the true covariation between N deposition and species richness will end up being residual variation because it cannot be explained by the modelled deposition estimates. Since this residual variation is likely to exhibit spatial structure, because N deposition varies spatially, it will contribute to spatially autocorrelated residual variation and therefore to the estimated non-independence attributable to unknown factors (*De Knegt et al., 2010*). The key difference between the P&J20 model and the MEA10 model is that the spatial structure in the residual error is modelled in the former. While P&J20 do this because they wish to correctly account for non-independence, a problem arises when this non-independence reduces the degrees of freedom available for estimating the mean error variance which increases as a result and thus reduces the apparent explanatory power of N deposition. Essentially, in the models of P&J20 the addition of an extremely flexible spatial field, with minimal cost in terms of number of parameters, accounts for a large proportion of the spatial structure that might otherwise be attributable to N deposition but for the measurement error. Thus the conclusion of P&J20 is only defensible if the spatially structured error is not attributable to N deposition. Their conclusion reflects the implicit yet critical assumption that it is not. The approach taken in MEA10 and SEA04 does not make this assumption. Below we illustrate the effect described above by fitting models to simulated data.

## A simulation study

To demonstrate the difficulty in interpreting spatial attribution models when covariates are measured with error, we conducted a small scale simulation study. The aim of this study is to examine how the spatial structure in the covariate and the addition of a flexible spatial random field can trade off against each other making inference problematic. We first simulated a hypothetical spatial covariate on the unit square with resolution $100 \times 100$.

This is simply a mathematical deterministic equation based on the x and y coordinates of the unit square given by the formula $\exp(2x - 1.2y^2)$ and therefore remains the same across each simulation run. A plot of this derived, hypothetical covariate is given in Fig. S1. For brevity, we have placed all graphical outputs associated with the simulation into the Supplemental Information.

Based on this spatial covariate a true, known response variable was established using the following relationship: $Response_i = 10 + 2 \cdot SpCov_i + \varepsilon_i$, where SpCov is the spatial covariate previously simulated and $\varepsilon \sim N(0, 0.5)$. The true relationship between our simulated covariate and response variable is shown in Fig. S2. To obtain a realistic dataset to model, we first drew a random sample of 100 observations from the full population of 10,000 values and then derived a new spatial covariate based on averaging the original across $20 \times 20$ blocks and adding in some random noise, we denote this SpCovErr. The relationship between the response variable and SpCovErr is shown in Fig. S3. It is clear that, despite the addition of random error and the averaging of the covariate, the true relationship (shown in Fig. S2) still holds. This pseudo-sample data was then analysed by fitting two separate models, both using the INLA (*Rue, Martino & Chopin, 2009*; http://www.r-inla.org) inference engine as used by P&J20. The first was a standard linear regression model including an intercept and the spatial covariate SpCovErr and the second model was the same linear regression model with the addition of a random spatial field. The random spatial field used the same flexible and computationally efficient SPDE approach as in P&J20. For both models the estimated effect size of the SpCovErr coefficient, the estimated intercept and the Deviance Information Criterion (DIC), used as a critical model comparison metric in P&J20, were stored. For the spatial models, the estimated spatial random field was also stored. This whole process was repeated 100 times simulating new hypothetical data each time and the resulting parameter estimates are shown in Fig. S4.

The results from the analyses of the simulated data show that when the random field is included in the models, the estimated effect attributable to the covariate diminishes, despite us knowing that this is a true effect. In only 18 of the 100 simulated analyses did we find a significant effect of the spatial covariate (based on the 95% credible interval), compared to all 100 for the simple linear regression. When comparing DIC values across the two model types, in all 100 cases the model with the spatial field had lowest DIC. As the DIC measures the goodness of fit of models penalised by the number of parameters and the lower DIC indicates a more parsimonious fit, in all cases the spatial field models appear to be more suitable. It is this same result that is used in P&J20 to suggest that spatial field models should be preferred to spatial covariate models, hence they suggest a lack of evidence of spatial covariate effects. Yet we know from how the data were simulated that this is not true and hence DIC is perhaps not an appropriate measure with which to make such conclusions. Further examination of the fitted random fields (Supp Fig A) demonstrated how this flexible surface had captured the spatial effect of the covariate which was lost when the covariate was averaged, thus mirroring the pattern shown in Fig. S1.

For comparison, the models were refitted across all 100 simulated data sets using the original SpCor covariate rather than SpCorErr. That is, we fitted all models using the covariate measured without error. Parameter estimates from these fitted models are shown

in Fig. S5. It is clear in this case that the addition of the spatial field has little influence on any inference we might make. Hence, when the implicit assumption that covariates are measured without error is met, the modelling and inference is robust to potential confounding of spatial attribution.

Our simulation shows that the assertion by P&J20 that spatial Bayesian linear models indicate small and ambiguous effects of N deposition on species richness is unreliable. We note that P&J20 acknowledged the possible importance of measurement error and in doing so highlighted the very problem that we explore. However, while they raised its possible importance they did not introduce it into their model specification. Our simulation shows that it is both fundamentally important and undermines their central claim.

P&J20 highlight the possibility that small sample size and selection on $P$ value thresholds can increase rather than decrease effect sizes as measurement error increases (*Loken & Gelman, 2017*). We believe this scenario is not relevant in this instance for the following reasons: The sample sizes in MEA10 were sufficiently large (number of 1 km sqrs = 239) such that attenuated slopes have a much greater chance of occurring than increased slopes (*Loken & Gelman, 2017*). SEA04 is admittedly less certain in this respect ($n = 68$ sites) but note that both studies link multiple species richness values to single $5 \times 5$ km N dep estimates hence the deviation between response and explanatory variable is an average of the within grid cell responses. Being an average this means that the fitted response is itself less likely to reflect random error that could increase rather than decrease the fitted slope at small sample size.

We agree that the issue of selection on $P$ values is a valid concern and reflects a much broader challenge to the global scientific community. We also agree with P&J20 that estimation of effect sizes and their uncertainty should be used more frequently to evaluate the ecological significance of a statistically significant result (e.g., *Smart et al., 2008*). In this respect it is a valid criticism of MEA10 that the omission of standardised effect sizes and their uncertainty made the results from that study less useful to researchers seeking to understand the magnitude of N deposition impacts.

## Evidence of measurement error in modelled N deposition

The flaw we highlight in P&J20's reasoning becomes more plausible if N deposition estimates at the $5 \times 5$ km square resolution indeed average out significant variation at finer resolutions across Britain. We show that this is the case by analysing the differences in variance of modelled N deposition estimates at the $1 \times 1$ km versus $5 \times 5$ km resolutions. We do this by comparing the variance of the 100 $1 \times 1$ km estimates versus the four $5 \times 5$ km estimates within each of the $10 \times 10$ km grid cells across Britain. This is not a direct comparison with the modelled estimates used in MEA10, SEA04 and P&J20 because equivalent $1 \times 1$ km estimates are not available. Here we use 1 km estimates from the FRAME model (*Dore et al., 2007*) for 2017 and simply average across the 25 1 km estimates in each $5 \times 5$ km grid square to produce a $5 \times 5$ km average. Plotting the $1 \times 1$ km versus $5 \times 5$ km estimates shows that substantial between-square variance at the 1 km resolution is not captured at $5 \times 5$ km (Fig. S6). This variance is mainly attributable to reduced dry deposition (NHx). This is not surprising because the main emission sources are farm
livestock whose density varies at the field and farm level and so is subject to considerable averaging as the resolution is coarsened (*RoTAP, 2012*). Thus, across Britain the 1 km/5 km model variance ratio is highest in the more agriculturally intensive areas –compare our Fig. 1 with Fig. 4.1 in *RoTAP (2012)*. In the 294 10 km squares within which the samples from MEA10 and SE104 were located, 45% had variance ratios greater than 2 (median 1.63, mean 4.95) showing that N deposition across the 1 km squares tended to be averaged at the 5 km resolution (Fig. 2).

Below, we comment on a number of other points raised by P&J20 of relevance to the detection of N deposition effects:

## Omitted variables and confounding with N deposition effects

P&J20 highlight the possible influence of unknown or known but unmeasured variables in large-scale attribution analyses. This is certainly a major challenge for such studies particularly those inferring a temporal response based on spatial patterns (*Damgaard, 2019*). We have explored these issues in our own work, for example carrying out a simulation analysis of the effect of omitted variables on the magnitude and sign of regression parameters (*Smart et al., 2012*). SEA04 attempted to account for confounding variables by careful site selection such that other variables were held as far as possible constant. MEA10 introduced other covariates and did not subset the data to achieve controlled or crossed driver gradients. Hence, both studies acknowledged but then handled the omitted variable issue in different ways. This reflects the different questions posed by each study. SEA04 were interested in optimising estimation of the effect of N deposition at the large scale while MEA10 were interested in whether an N deposition effect was detectable across British acid grasslands in the realistic presence of other potential causes of species richness difference including human activity. We agree that both studies would have benefited from a more in-depth exploration of the influence and causes of spatially structured non-independence in the species richness responses after fitting covariates. The key test that is required is to refit the models of P&J20 using more accurate measurements of N deposition at the location of each species richness observation. Without such a test we cannot unequivocally demonstrate that a fraction of the residual spatial autocorrelation that was assigned to the random field is indeed explainable by N deposition. However, we have shown that this is a plausible scenario. This being the case the conclusion of P&J20 is not proven by their analysis.

## Experiments versus spatial gradient studies for detecting N deposition effects

P&J20 cite two sets of experimental results as support for their conclusions. They refer to the absence of significant changes in species richness in *Phoenix et al. (2012)* as supportive but then criticise these as being based on *P* values leaving the reader to wonder whether they believe in them or not. They also find support in the results from the long-running Rothamsted Park Grass experiment (*Storkey et al., 2015*) but point out that the species richness trends may not be reliable because they are confounded with area and treatment changes.

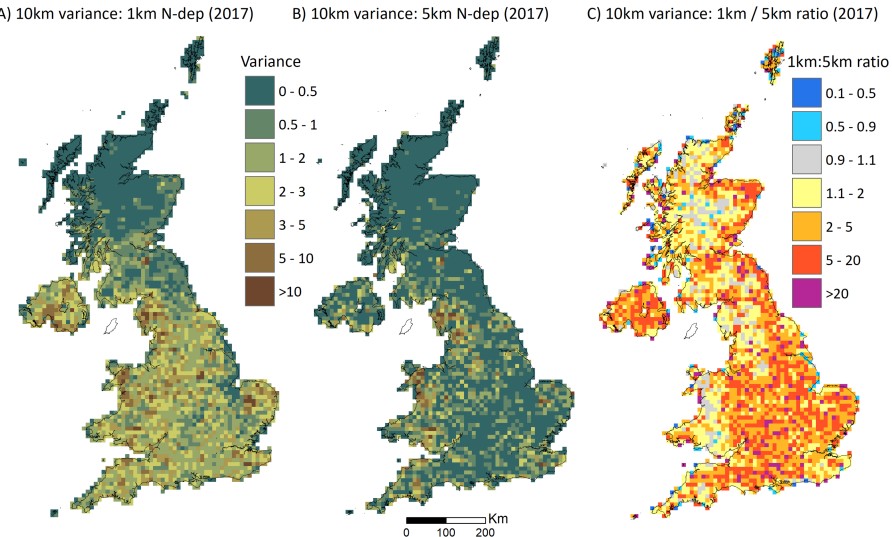

**Figure 1 Variation in modelled nitrogen deposition (N-dep) across Britain.** A ratio of 5 indicates that the variance in modelled deposition between the 1 km squares within a 10 km square is five times that of the variation between the 5 km squares within the 10 km square.

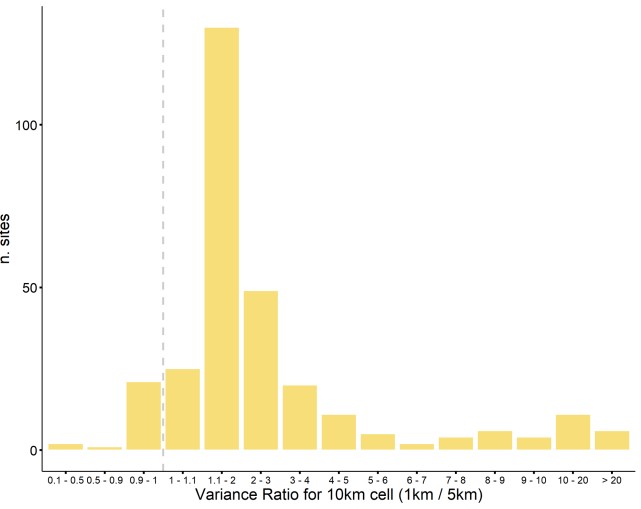

**Figure 2 Variance ratios for 10km squares containing samples from MEA10 and SEA04.**

We further question the support claimed from *Phoenix et al. (2012)*. Unless experiments have been running for a significant fraction of the 200 years over which N deposition increased across the British landscape (*Fowler et al., 2000*) they cannot directly quantify the vegetation dynamics that have resulted from long-term cumulative deposition effects. Likewise, relatively short-duration experiments cannot be used to explore the loss of diversity that may have occurred many decades prior to the start of the experiment despite these patterns persisting into the present and being detectable across contemporary

spatial deposition gradients. These points are made clearly by *Phoenix et al. (2012)* as possible explanations for the lack of species richness response in their experiments. So both studies appear to have shortcomings in offering support for the conclusion of P&J20 that "*… experimental data using realistic applications of Ndep appear to support our finding that richness is a relatively insensitive metric of such impacts* ". The plurality of supportive independent results implied by this statement boils down to just two at both of which P&J20 level criticisms.

Lastly, readers should be aware that the vegetation in the control plots at Rothamsted sample one type of neutral grassland now very rare in the British countryside (*Dodd et al., 1994*); so rare in fact that it is a Priority Habitat under current British legislation. Hence, while P&J20 state that the control plots "*may be a useful comparator for some habitats in the wider landscape*" the comparative role envisaged is likely to be limited.

A way to use experimental results to arbitrate between the models of P&J20 would have been to compare the observed species richness in control plots, for example from the two acid grassland studies in *Phoenix et al. (2012)*, with model predictions. This would help determine whether the cumulative consequence of long-term trends in deposition has resulted in values of species richness that are consistent with any of P&J20′ s competing models. Benchmarking experimental baselines in this way could also help manage expectations for future responses to treatment effects (e.g., *Hanson & Walker, 2020*).

## Concluding remarks

We fully subscribe to the need for transparency in reanalysing and debating contested scientific results. As P&J20 point out this is especially important when evidence has a major influence on policy and public expenditure on research. However the conclusions from their analyses are undermined by not exploring the effect of measurement error within their model specification.

Finally, we thank P&J20 for their in-depth reanalysis of our data and reappraisal of our results. It has alerted us as to how spatially structured residual variation can highlight not just omitted variables but omitted variation associated with a focal causal variable yet where this variation is not quantified by the covariate selected to represent the causal agent. This is likely to arise when process-based modelled estimates are used as covariates and where the estimate is a grid cell average and therefore cannot explain the variation in a more finely resolved sub-grid response. We also agree with P&J20 that the most important outcome of studies seeking to attribute signals to a range of potential causal factors is an estimate of effect size. However, the effect size attributed to N deposition impacts in large observational studies will vary depending on the inclusion of other covariates. In this respect experiments have a contributing role to play in providing benchmark estimates of the potency of N deposition in driving change when other factors are controlled.

## Data supplied by the author

An R script for the simulation analysis and associated figures are available as Supplemental Files.

### Funding
The research was supported by the UK-SCaPE program delivering National Capability (NE/R016429/1) funded by the Natural Environment Research Council. The funders had no role in study design, data collection and analysis, decision to publish, or preparation of the manuscript.

### Competing Interests
The authors declare there are no competing interests.

### Author Contributions
- Simon M. Smart conceived and designed the experiments, performed the experiments, analyzed the data, authored or reviewed drafts of the paper, and approved the final draft.
- Carly J. Stevens and Lindsay C. Maskell conceived and designed the experiments, authored or reviewed drafts of the paper, and approved the final draft.
- Sam J. Tomlinson conceived and designed the experiments, performed the experiments, analyzed the data, prepared figures and/or tables, authored or reviewed drafts of the paper, and approved the final draft.
- Peter A. Henrys conceived and designed the experiments, performed the experiments, analyzed the data, prepared figures and/or tables, authored or reviewed drafts of the paper, and approved the final draft.

### Data Availability
  Code supporting our analysis is available in the Supplementary Files.

### Supplemental Information
Supplemental information for this article can be found online at http://dx.doi.org/10.7717/peerj.10632#supplemental-information.

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
