# Peer review of "Comment on Pescott & Jitlal 2020: Failure to account for measurement error undermines their conclusion of a weak impact of nitrogen deposition on plant species richness"

_PeerJ, doi:10.7717/peerj.10632_

## Round 0.1 · original submission · Major Revisions

Dear Authors

Two in-depth reviews of your manuscript have been received. It is my opinion that this kind of scientific discussion in the academy is enriching.
Given that the author of the paper you are commenting on has reviewed the paper I think you have an opportunity to improve your manuscript taking into account his comments.

If you feel that you can reply and implement his comments, as well as those from the other reviewer I will be willing to reconsider your manuscript for publication.

Best regards

·

Basic reporting

In general the English is clear, unambiguous and professional throughout. In a couple of areas I find the language problematic. For example: Line 20 the authors refer to Pescott & Jitlal (2020) methods as "sophisticated" and then use the term "simple" in relation to their simulation (Line 25). I find this distinction odd as both use the same types of models (one is simpler than the other but this gives the impression of suggesting that the sophistication is a contributing factor to the perceived "premature conclusions" of Pescott & Jitlal).

In Line 240, 247 and 252 the authors use the terms "their" and "our" but it is not clear who they or our are - I did not read the original paper as a direct criticism of an individual's/group's work but rather as an important consideration that all researchers in this (and other observational sciences) need to take account of (spatial autocorrelation).

After Line 165 the paper begins to read like a point-by-point rebuttal rather than a comment highlighting concerns with the interpretation of results from P&J20. Maybe structuring the comment around the main issue that are raised rather than every perceived divergence between the two groups of authors would be beneficial to the reader.

Figures - there are far too many figures to go in to the main document with legends that are not informative enough. What, for example, is the need for both Figure 2 and 3 in the main document? I would be tempted to move all the Figures in to supplementary materials except Figure 7.

Code - It is great that the code is shared.

Experimental design

The simulation design appears sound. I do have one question - the authors say that the DIC is lower in the models which include the spatial field (Line 134 to 141). I am not sure I understand this point (I am not an expert in this at all). I assume a model with a lower DIC has a higher predictive accuracy. Have I got this the wrong way around? - perhaps they can make this paragraph clearer (idiot proof!).

Validity of the findings

I have a couple of areas where I question the presentation of the findings rather than the validity (I hope it is okay for me to use this section for those). The premise of this comment is that the authors believe (and provide lines of evidence) that the findings of P&J20 are premature in that they suggest caution in interpreting observational assessments of the effect of N deposition on biodiversity. The main focus of the comment is on the failure to account for measurement error. This is a valid concern which has been highlighted in epidemiologi recently. I find it strange however, that the two sets of authors (who appear to work in the same organisation) have not joined forces on a paper using the datasets that are presented here and in the original to actually assess the effects of measurement error in combination with spatial autocorrelation. The authors state that (in Line 198) that the key test is to refit the P&J20 models with more accurate measurements of N deposition. Why has this assessment not been done? Surely this would be much more valuable for the understanding of the true effects of N deposition.
The last paragraph (Line 263 - 268) suggests that the authors feel that this is a valid point - that a "better" assessment of the causal effects of N on biodiversity are needed urgently.

I find the repeated assertation that the conclusions of P&J20 should be treated as doubtful and not proven could also equally be applied to this comment - the weight of evidence put forward is circumstantial and does not unequivocally prove that the findings of P&J20 are incorrect. I would have preferred to see an actual assessment of the data taking in to account the measurement uncertainty which is demonstrated in this comment. What are the reasons why this has not been done - is it that the data are not available (if yes - then this should be highlighted here as an important next step), or is there another reason(s)?

Additional comments

Line 81 states that the original authors overlook measurement error but line 149 states that they do address it - they dont include it in the model and suggest a reason for this. These two statements need to be reconciled.

Line 51 states that causal modelling in ecology is tough (yes it is!). The authors highlight the problems around these but then disagree with P&J20 who cast doubt on the size of the effect of N on biodiversity previously reported after trying to empirically account for some of these problems. Do the authors disagree that prior modelling of the effect of N deposition on biodiversity has been problematic (the true effect has not yet been determined) or that the true effect has been determined and this is reflected by previous work despite the problems that they highlight. This is not clear throughout the comment and I think a clearer statement is needed.

·

Basic reporting

See below.

Experimental design

See below.

Validity of the findings

See below.

Additional comments

Note that because the paper is a response to a paper that I authored, I do not provide a full response to the substantive points of Smart et al below, I only comment on aspects where it seems to me that the authors misrepresent Pescott & Jitlal (2020). I do this because I may respond to the interesting and substantive aspects of Smart et al. in an actual paper. I have just selected "minor revisions" in order to submit my review. However, as I am reviewing a rebuttal, I leave it to the editor and other reviewers to make decisions about revisions, publication etc.

Given the situation I leave it to the editor and the authors to decide whether they wish to make any amendments in response to the below points given, but I may comment directly on any article paragraphs that are misleading once the article is published if they are not altered. For these reasons, I also see no point in my re-reviewing whatever revision is submitted.

Taking the above into consideration, the below points are by line number in the PeerJ reviewing PDF.

##Overal conclusion
Smart et al. provide a useful example of a situation where uncertainty and error in space-for-time Ndep impacts modelling can mask the truth. However, their model merely portrays one possibilty, and many others are possible, including situations where unmeasured confounders lead to incorrect conclusons, and where finer-scale Ndep estimates still support a conclusion of small impacts. Moreover, fine-scale Ndep estimates would still require measurement error modelling (predictions from the FRAME model at 1km are obviously not the truth), so it seems likely that estimated coefficients will still exhibit important uncertainty in such situations. Overall then, the work of Smart et al. supports one of the key points of Pescott & Jitlal, that clearly and accurately reporting impact estimates with all associated uncertainty (including that which is not covered by model outputs, but which is related to having chosen one model out of numerous possible models in the first place) is essential to move this field forwards. In my opinion Smart et al. could make more of this point, acknowledging the many issues with much of the previous work in this field that P&J raise, rather than just putting objections to one aspect of P&J front and centre in their piece, and using this language to insinuate that everything about P&J is somehow unreliable. The lack of any discussion of estimated effect sizes, whether with or without the spatial field (which doesn’t actually make that much substantive difference) is also conspicuous in respect of this.

#Title/Abstract
This is somewhat misleading. P&J said that the modelling approach they had taken indicated small and ambiguous effects, and that previous estimates were likely to be have been overestimated etc. Reading this title and the abstract you would imagine that P&J were absolute in their conclusions, and that these conclusions were stated to be independent of the models they used.

L19 It’s interesting that Smart et al. refer to false positive results here, whereas the entirety of P&J was about effect sizes and uncertainty. I just note this in passing as an example of how pervasive the focus on P-values often is.

L26 The simulation study presented by Smart et al. is useful and interesting, but it is a simulation of a possibility, not the truth, so it seems odd to say that the conclusion of P&J is flawed. A more accurate and less rhetoric-laden statement would be that Smart et al. have demonstrated one situation in which the model used by P&J may be misleading.

It is also misleading that Smart et al. state that P&J “failed to recognise” that covariate measurement error could affect our results. What we actually said was:

“Other uncertainties relating to our conclusions pertain to the fact that the broad-scale spatial field used here may be accounting for information that, if known, would change the size of the Ndep impact regression coefficient. This could be in the form of additional covariates, or more highly resolved estimates of the Ndep load that a location has actually received. The Ndep estimates used here (and by the original studies) are resolved to a grain size of 5 x 5 km, and this additional uncertainty could have attenuated our estimate of the regression coefficient (the measurement error in explanatory variables problem; e.g., see Fox, 2016), even if there is no systematic bias relating to the association of 5 x 5 km-estimated Ndep levels with particular types of vegetation.”

Obviously we did not explore this in detail, but it is inaccurate to say that we completely failed to recognise this issue.

L38 I agree that the conclusions of P&J were “not proven”. Can Smart et al. point to an observational study using inductive logic that is absolutely proven and not merely a point on a journey to fuller understanding? Do they perhaps consider that their earlier studies on Ndep are all proven and that none were “premature”, whatever that means? More rhetoric.

L59 This is confusing, as you use the word “significance” vaguely, which conflates P-value threshold-based conclusions with the normal usage of the word. It’s also worth noting the analyses of P&J that exactly replicated the model of MEA10 (i.e. the one without the spatial field) did not exactly show a large effect size, so some elaboration on how this statement about our conclusion of “weak effects” actually differs from MEA10 would be useful. After all, our main criticism of MEA10 was the failure to report any effect sizes from the multivariable model run (and, indeed, the replacement of these with a plot featuring a univariable regression instead).

L62 No, the main conclusion is that space-for-time studies of Ndep have either not used appropriate models and/or have not reported effect sizes; spatial auto-correlation was part of this but was not the only or main conclusion. I would say that the main points of our paper were about whether or not it is worth collecting more of these type of data and running these types of studies, and about better reporting standards (uncertainty, effect sizes, not relying on P-value thresholds to detect effects, acknowledging model-dependency of results clearly).

L64 Given that P&J conclude by saying (in their abstract) “we suggest that a greater focus on clearly reporting important outcomes with associated uncertainty, the use of techniques to account for spatial autocorrelation, and a clearer focus on the aims of a study, whether explanatory or predictive, are all required.” perhaps Smart et al. could clearly state what they mean by “the central conclusion of P&J”.

L71-74 Overall this is misleading, because, for SEA04, the inclusion of the spatial field was not the only difference. If you are focusing on spatial autocorrelation, please clearly separate this from the issues of P-value-based SVS exhibited by SEA04 (and also the lack of random effects, and the wrong support that used for the model, i.e. Gaussian rather than Poisson or some other distribution suitable for count data). As with an earlier point above, it would also be useful to have more clarity around what a “weak effect” (L66) is in relation to the issues with spatial autocorrelation. Without a spatial field P&J estimated a median effect for MEA10 of -2%, which would have been reported by MEA10 if they had reported any effect sizes from their multivariable model. The same estimate for SEA04 (vascular plants, all variables, monad RE only) is -1.5% (this can be verified from our R code, although we did not present this result in our paper). (Recall that this is compared to -1% under the model with the spatial field). Our point here is not that the model of Smart et al. is not a possibilty, but that even the models without the spatial field do not indicate what we would consider to be large effects. This means, of course, that the criticisms here also apply to SEA04 and MEA10; some clearer acknowledgement of this would be valuable, rather than just stating throughout that P&J are somehow wrong in isolation.

L77 Not true. Our paper gives numerous reasons for reanalysing the data of SEA04 and MEA10, the central reason and initial motivation was just to get an effect size that was absent from the literature after 16 years+ of research on this topic!

L81 Not sure how we overlook it given that we discuss it in the Discussion. See the quote above.

L104 Not sure how this was implicit given that we discussed it.

L104/105 Again, interesting how you elide SEA04 and MEA10 here even though they used quite different approaches. The effect of this statement is to overlook the issues with PSVS and implicitly endorse that approach. Do you?

L150/151 This is misrepresentation. We do not write it off as “too difficult to assess”, what we said was (feel free to quote us) was:
“Whether or not the potential for these biases is more serious for inference than the absence of covariates that are unavailable, such as historic land management events (e.g., Rackham, 1986), is difficult to say.”

The simulation conducted by Smart et al. applies to one particular model of reality, we do not know whether that model is true in respect of Ndep impacts on richness. Other simulations involving unmeasured confounders with different types of spatial structure are possible (e.g. historic land management events etc.) What we essentially say in our quote is that it is difficult to know the truth of the relative importance of measurement error in independent variables and unmeasured confounder variables. It would be easy for me to say, in line wth L111 of Smart et al., that the assumptions of Smart et al. could be false in this respect.

L180 What proportion of the SEA04/MEA10 samples are actually in these high ratio areas?

L189 Maybe, but they didn’t assess whether this was successful, or consider the impacts of it not being completely successful on their inferential strategy.

L192 Maybe, but no-one in the statistics literature seems to think that it is a good idea to use PSVS to do this. See some of our references.

L241 is it a “likely” flaw or a “possible” flaw? At the end of day Smart et al. have asserted that if we had access to better estimates of Ndep at the sites (and there really are no important unmeasured confounders) we would get larger effect sizes; we still do not know that this is the case. Moreover, there is still the issue of whether 1-2% drops in richness are to be considered a large or small effect.

L254 I don’t know whether Cramer et al.’s concept of “detection and attribution” relates to P-value cut-offs, but it would be good to be clear about what the aims of the area of space-for-time Ndep studies actually are. To me quantifying effects and their variation is the most illuminating thing to do; merely “detecting” an effect at some P-value cut-off is not as useful in my opinion.

L246 Rather than airily dimissing unspecified “philosophical points”, it would be more useful to discuss these issues. All the points made by P&J have been raised as important for actual estimates of effect sizes from observational data (this is why they were included and clearly referenced), and some of them may be particularly relevant for attempts to make causal inferences about effects from space-for-time data (e.g. proxy variables).

L259 Are they potentially flawed or flawed? In other places you just say that they were outright flawed.

L261 Again, hard to know what Smart et al. consider “importance” to mean here. All of the effects we reported were in the 1-2% loss of richness ballpark (even if I run a Poisson GLMM without the mesh or 1km square RE on the SEA04 data with Ndep as the only covariate the effect is only -2.2%). How does this relate to the effects of management, climate change etc.? How do these effect sizes relate to the “widespread loss” claimed by MEA10? Or the “linear function” claimed in the abstract of SEA04? One gets the impression that effect sizes, or partial effects, have been ignored in this field, and “significance” focused on, precisely because the detectable effects are relatively small.

---

## Round 0.2 · Minor Revisions

Dear Authors

Please, implement the minor revisions indicated by the reviewer. Many thanks!

·

Basic reporting

This is a much improved manuscript and most of my concerns have been addressed. I have enjoyed being "on the side-lines" of this debate that appears to be far from settled. The article is much clearer and I appreciate the effort the authors have made to clarify their language and to add explanations where needed.

I find the section "other points raised" (line 272 - 283 in the marked-up word document) redundant. I think the ms gains nothing from this paragraph. The authors do not address the philosophical points raised in P&J here (which is that stated aim of the paragraph) but rather just state that they are qualified scientists who understand the study system (which is evident from the rest of the ms and from their other publications which they refer to). Removing this entire paragraph will improve the ms greatly and I urge the authors to do so.

Experimental design

The simulation study is repeatable using the code provided and the methodology is sound.

Validity of the findings

This version of the ms reads much less like a point-by-point rebuttal and highlights some key points for debate and objectives for future research in this field.

Additional comments

Line 37 in the abstract needs a comma I think: "Until then, the conclusions of..."
Also commas need here:
Line 82: "However, ..."
Line 118: "For brevity, we..."
Line 201: "Thus, across Britain..."
Line 216: "Hence,..."
Line 289: "However,..."


Comment on style:
Lines 210-213: Three consecutive sentences start with "We" - maybe reorder or reword these to make it more readable.

Line 303: Delete "be" in the sentence: "...in large observational studies will vary depending on the inclusion of other..."

---

## Round 0.3 · accepted · Accept

Dear Dr. Smart,

Thank you for your submission to PeerJ.

I am writing to inform you that your manuscript - "Comment on Pescott & Jitlal 2020: Failure to account for measurement error undermines their conclusion of a weak impact of nitrogen deposition on plant species richness" - has been Accepted for publication. Congratulations!

·

Basic reporting

Great - my comments and suggestions have been addressed

Experimental design

No comment

Validity of the findings

No comment